# Transcriptional Systems Vaccinology Approaches for Vaccine Adjuvant Profiling

**DOI:** 10.3390/vaccines13010033

**Published:** 2025-01-01

**Authors:** Diogo Pellegrina, Heather L. Wilson, George K. Mutwiri, Mohamed Helmy

**Affiliations:** 1Vaccine and Infectious Diseases Organization (VIDO), University of Saskatchewan, Saskatoon, SK S7N 5A2, Canada; diogo.pellegrina@usask.ca (D.P.); heather.wilson@usask.ca (H.L.W.); george.mutwiri@usask.ca (G.K.M.); 2Vaccinology and Immunotherapeutics Program, School of Public Health, University of Saskatchewan, Saskatoon, SK S7N 5A2, Canada; 3Department of Veterinary Microbiology, Western College of Veterinary Medicine, University of Saskatchewan, Saskatoon, SK S7N 5A2, Canada; 4Department of Computer Science, University of Saskatchewan, Saskatoon, SK S7N 5A2, Canada; 5Department of Computer Science, Lakehead University, Thunder Bay, ON P7B 5E1, Canada; 6Department of Computer Science, Idaho State University, Pocatello, ID 83209, USA; 7Bioinformatics Institute (BII), Agency for Science, Technology and Research (A*STAR), Singapore 138632, Singapore

**Keywords:** adjuvants, vaccines, systems vaccinology, transcriptomics, bioinformatics

## Abstract

Adjuvants are a diverse group of substances that can be added to vaccines to enhance antigen-specific immune responses and improve vaccine efficacy. The first adjuvants, discovered almost a century ago, were soluble crystals of aluminium salts. Over the following decades, oil emulsions, vesicles, oligodeoxynucleotides, viral capsids, and other complex organic structures have been shown to have adjuvant potential. However, the detailed mechanisms of how adjuvants enhance immune responses remain poorly understood and may be a barrier that reduces the rational selection of vaccine components. Previous studies on mechanisms of action of adjuvants have focused on how they activate innate immune responses, including the regulation of cell recruitment and activation, cytokine/chemokine production, and the regulation of some “immune” genes. This approach provides a narrow perspective on the complex events involved in how adjuvants modulate antigen-specific immune responses. A comprehensive and efficient way to investigate the molecular mechanism of action for adjuvants is to utilize systems biology approaches such as transcriptomics in so-called “systems vaccinology” analysis. While other molecular biology methods can verify if one or few genes are differentially regulated in response to vaccination, systems vaccinology provides a more comprehensive picture by simultaneously identifying the hundreds or thousands of genes that interact with complex networks in response to a vaccine. Transcriptomics tools such as RNA sequencing (RNA-Seq) allow us to simultaneously quantify the expression of practically all expressed genes, making it possible to make inferences that are only possible when considering the system as a whole. Here, we review some of the challenges in adjuvant studies, such as predicting adjuvant activity and toxicity when administered alone or in combination with antigens, or classifying adjuvants in groups with similar properties, while underscoring the significance of transcriptomics in systems vaccinology approaches to propel vaccine development forward.

## 1. Introduction

Adjuvants are substances that are added to vaccines to enhance the body’s immune response to the co-administered antigen, which itself is not sufficiently immunogenic to elicit optimal immune responses. They are especially critical for subunit vaccines, as these proteins alone might not be recognized as a threat by the immune system, but when co-administered with adjuvants induce the proinflammatory signals that activate and recruit immune cells to the injection site [1]. For almost a century, adjuvants have played a key role in improving the efficacy of vaccines by stimulating the immune system to respond more vigorously to vaccine antigens. The first licensed vaccines with adjuvants were produced in 1930, with the addition of aluminium salts (Alum) to the diphtheria and tetanus vaccines. Although Alum was added to the formulation of several other vaccines, no other type of adjuvant was licensed for use until a monophosphoryl lipid A (MPA) adjuvanted the human papillomavirus (HPV) vaccine in 2009 [2,3]. Since then, several other adjuvants have been developed and, according to FDA data, about 35% of currently commercially available vaccines in the United States are adjuvanted [4].

Despite the importance and wide-spread use of adjuvants in vaccine formulations, the mechanisms that they use to enhance immune responses are still not fully understood, and remain an active area of research [5]. It has been demonstrated that some adjuvants direct and amplify specific adaptive immune responses by engaging innate immune cells and activating signaling pathways via pattern recognition receptors (PRRs) [6], while others are very effective at increasing antigen availability to B cells in draining lymph nodes, where they are activated and go through the processes of affinity maturation [7]. In some cases, adjuvants are necessary for the vaccine to stimulate and then maintain cells that produce antibodies against the desired antigens. In other cases, certain adjuvants have been shown to reduce the amount of antigen needed, lower the number of doses required, and improve the longevity of the immune responses induced by vaccines [8]. The size of the adjuvant particles or vesicles is also important, as 200 nm is the largest size any particle can diffuse freely into the lymph nodes, while particles smaller than 50 nm may leave the lymph node too quickly [9]. The lymph node is a particularly important tissue as it is where the antigen is taken by dendritic cells to be presented to T cells and B cells.

The application of systems biology approaches in vaccine research and development is known as “systems vaccinology”. Such an approach is used to provide insights into the complex and dynamic interactions between vaccine components (adjuvants and antigens) and host cells. Biologic changes resulting from these interactions may provide a deeper understanding of adjuvant activity in a holistic and mechanistic manner, such as understand how adjuvants influence several aspects of the immune system like the activation of antigen-presenting cells (APCs), T-cell regulation, or tumor immunology [10,11].

Adjuvant activity can be impacted by antigen dose and composition, the route of immunization, and the species to which the vaccine is being administered. In a study that compared immune responses to an antigen when formulated with panels of adjuvants, the responses observed indicated a few pathways being enriched with differentially expressed genes (DEGs) in all adjuvants, while other pathways were exclusive to some adjuvants [12]. Another study [13] assessed gene expression in mice when injected with a trivalent influenza vaccine adjuvanted with MF59 (which comprises a squalene emulsion, surfactants, and a buffer solution), and compared the effects against mice injected with the same antigens but only components of MF59 ingredients. They demonstrated how multiple molecules must mutually interact to produce an adjuvant that induces a substantial immune response.

With the interest of better understanding the mechanism of action (MOA) of vaccine adjuvants, we reviewed previous studies describing how vaccine adjuvants alone or when co-administered with antigens interact with the immune system. We gave priority to transcriptomics in systems vaccinology approaches, as this approach provides information regarding the interaction between the vaccine and/or adjuvant components with all genes expressed in the sampled cell types. The profiling of this large number of genes, including those not canonically associated with the immune response, are fundamental since the complex regulatory networks responsible for immune response make it insufficient to understand the function of any one gene in isolation. This approach to understanding adjuvants’ mechanisms of action is required to achieve the rational selection of adjuvants in the process of vaccine formulation.

To maximize the comprehensiveness of our survey, we searched the Gene Expression Omnibus (GEO) [14] and Google Scholar for the keywords “vaccine”, “adjuvant”, and “transcriptomics”. From the search results, we curated 23 papers that used transcriptional systems vaccinology approaches to investigate the mechanisms of adjuvants, or that compared the effects of different adjuvants. The selected works had experimental designs that clearly compared the application of adjuvants against a negative control, appropriately presented both their methods and their results, and were relevant for our purposes. Table 1 overviews the 23 studies that matched these criteria and shows how they are varied regarding experimental methods, data availability, and the choice of adjuvant and antigen.

## 2. Adjuvants as Enhancers of Vaccine Efficacy

Adjuvants can help in the formation of long-lasting immune memory, ensuring that the host immune system can recognize and respond quickly and with greater strength to similar pathogens upon subsequent encounters with similar antigens or pathogens. Since adjuvants have complex interactions with various antigens and the host organism, each adjuvant may only be suitable to a subset of vaccines. By enhancing both humoral and cellular immunity, adjuvants help generate a more robust and comprehensive immune defence that may be vital for protection against rapidly evolving pathogens with high genetic variability. The adjuvant MF59, for instance, has been shown to improve protection in subsequent influenza seasons, while the adjuvant AS04, used in the HPV vaccine Cervarix, has been shown to elicit strong antibody responses, as well as robust cell-mediated immunity, providing cross-protection against multiple HPV strains [35].

By enhancing the immune response, adjuvants like MF59 can allow for a reduced amount of antigen in each dose while achieving the same level of immunity [36]. This dose-sparing, or antigen-sparing, effect can be particularly important in situations where antigen supply is limited or costly, making it possible to produce more vaccine doses from a limited amount of antigen.

## 3. Adjuvant Classification

The design or selection of an appropriate adjuvant is crucial in the process of vaccine formulation and manufacturing. Thus, several approaches have been used to classify adjuvants to help facilitate the adjuvant design/selection process.

The Vaxjo database of vaccine adjuvants [37] classified 95 unique adjuvants that were tested in 415 unique vaccines, 290 of which contained subunit antigens. The database was made with automated annotation produced by text-mining adjuvant-related keywords from more than 35,000 articles published since 1948. The knowledge obtained using this data-mining approach is represented by Vaccine Ontologies (VO), which define terms and relations that logically represent biological entities and how they relate to each other, and are stored in a way that is easily interpretable both by human and machines [38].

Adjuvants in Vaxjo are classified into nine different classes. The three main adjuvant classes are mineral salts, emulsions, and microorganism-derived. Other adjuvant classes with less members include synthetic, cytokines, tensioactive, particulate, carbohydrates, and combinations. This classification is based on the chemical properties and the origin of the main immunostimulant ingredient of each adjuvant and provides useful information that could help in the adjuvant selection process. The Vaxjo database also provides information about antigen type, interactions with pathogens (if available), the host species, and the vaccine type. However, the Vaxjo database has not been updated for over a decade and several newer vaccines and adjuvants are absent [37].

A classification that could group adjuvants by their mode of action would be ideal for the adjuvant selection process. This approach was attempted by the Vaccine Adjuvant Compendium [39], which describes 83 vaccine adjuvants and classifies them according to whether their immune profiles involve Th1-, Th2-, or Th17-type T-cell responses or any mixture of these. The resource was launched in 2021 by the Division of Allergy, Immunology, and Transplantation (DAIT) and the National Institute of Allergy and Infectious Diseases (NIAID) at the NIH, but its list of adjuvants is noticeably incomplete, lacking most adjuvants licensed for human use, including AS03, AS04, MF59, and rVSV.

Although no consensus was reached to classify all discovered and theoretically predicted adjuvants, it is an easier task to classify the shorter list of adjuvants licensed for human use, a subgroup that is much more studied and cited due to its greater economic potential. According to the FDA’s list of licensed vaccines, twenty-one different vaccines use adjuvants that include some kind of aluminium salt (AS); seven use oily adjuvants, such those that include monophosphoryl lipid A (MPL) from the AS adjuvant family (the shingles vaccine (SHINGRIX) uses AS01B, the respiratory syncytial virus vaccine (AREXVY) uses AS01E, and the human papillomavirus vaccine (CERVARIX) uses AS04) or squalene emulsions (Arepanrix H5N1 uses AS03, and Fluad, Fluad Quadrivalent, and AUDENZ use MF59); and two include CpG oligodeoxynucleotides (HEPLISAV-B uses CpG 1018 and CYFENDUS uses CpG 7909). Table 2 shows the full list of adjuvanted FDA-approved vaccines.

## 4. Systems Vaccinology and Adjuvant Informatics Research

Investigations that simultaneously analyse a single gene or a small number of genes fail to properly characterise how adjuvants affect different pathways in different cell types. With systems vaccinology approaches, such as microarrays or RNA-Seq transcriptomics, the expression of thousands of different genes can be quantified pre- and post-administration. It is possible to catalog the many pathways involved in adjuvanted immune response through identifying DEGs. However, this approach fails to elucidate how these DEGs are distributed into different cell types that interact with each other, despite partial success using mRNA for the deconvolution of bulk cells into immune cell types [41].

Single-cell RNA sequencing (scRNA-Seq) is a modern technique that allows for the analysis of the transcriptomic content of individual cells, offering unprecedented resolution into cellular heterogeneity and complex biological systems [42]. In a scRNAseq analysis of mice injected with yellow fever virus vaccine adjuvanted by a TLR7/8 agonist, 3M-052, the authors of [29] observed two distinct monocyte clusters characterized by Ly6c2 and Ccr2 expression that regulated the genes involved in inflammatory response, such as the innate immune chemokines Isg15 and Cxcl10, as well as Cd74, a member of the MHC class II complex. It would be ideal to observe the effect of adjuvants in different cell types, but we were unable to find scRNA-Seq datasets that included more than one adjuvant (Table 1).

A similar result to what would be achieved by scRNA-Seq was obtained by [24], which used a modified experimental design to observe the effect of different cell types on the limitations imposed by bulk RNA-Seq. Mice were immunized subcutaneously using Plasmodium vivax circumsporozoite recombinant protein (PvCSP) with PolyI:C and Montanide ISA 720. After collecting cells from each harvested mice spleen, cells were sorted using flow cytometry and separated into CD4+ T cells, CD8+ T cells, and B cells, and the remaining cells were discarded. These three cell types were subjected to bulk RNA sequencing separately to discern how these expression profiles were impacted by the adjuvants. Both adjuvants enriched the DEGs in TNF-α signaling by the NF-κB pathway in both CD4+ and CD8+ T cells, and Montenide enriched the heme biosynthesis pathway in B cells. If it was not for the cell sorting step, the heme biosynthesis enrichment could be attributed to erythrocyte development, but in the context of B cells, heme is associated with the development of plasma cells through Bach2 inactivation [43].

Hence, we notice that transcriptional systems vaccinology approaches research has been employed for a limited number of vaccines and an even smaller number of adjuvants, but answers several important questions related to the following:Adjuvant profiling (comprehensive gene expression analysis of adjuvants): Bulk or single-cell transcriptomics provide a detailed profile of gene expression changes in response to adjuvant administration, offering a comprehensive view of the molecular events triggered by adjuvants [28,29,30]. By examining the entire transcriptome, researchers can identify which genes are upregulated or downregulated, uncovering the specific pathways and cellular responses activated by adjuvants. For instance, transcriptomic studies have revealed that the adjuvant MF59 induces a broad range of immune-related genes, highlighting its ability to activate multiple immune pathways simultaneously [44].Identification of biomarkers: One of the most significant advantages of employing transcriptomics in adjuvant research is the ability to identify biomarkers associated with immune responses. These biomarkers can serve as indicators of vaccine efficacy and safety and facilitate the design of more effective vaccines [45]. An example of this is the use of gene expression microarrays for the identification of a gene expression signature involving interferon-stimulated genes (ISGs) in response to the yellow fever vaccine YF-17D, which has been shown to predict the magnitude of the immune response and long-term immunity [46]. Identifying biomarkers of adjuvants will save time and costs associated with the laborious and cumbersome animal trials that are traditionally employed in the screening and testing of potential adjuvants.Understanding immune pathways: Transcriptomic data can reveal the activation of specific immune pathways and the roles of various immune cells, enhancing our understanding of how adjuvants modulate the immune system. By mapping the gene expression changes induced by adjuvants, researchers can pinpoint the impact on the signaling pathways involved in the immune response. For example, transcriptomic studies have shown that certain adjuvants activate toll-like receptor (TLR) signaling pathways, leading to the production of pro-inflammatory cytokines and the activation of adaptive immune responses [47].Facilitating the rational selection of adjuvants: Transcriptomics allows for the comparison of different adjuvant profiles (effects on gene expression), facilitating the rational selection of the most effective adjuvant for a particular vaccine. By comparing the transcriptomic profiles induced by various adjuvants, researchers can identify the molecular signatures associated with optimal immune responses. This comparative approach helps in selecting adjuvants that produce the desired immune outcomes, thereby enhancing the overall efficacy of vaccines. For example, a comparative transcriptomic study [12] has demonstrated that SAA3, an upstream regulator of Il1 in mice, is downregulated in mouse lymph nodes when injected with Engerix B (recombinant hepatitis B antigen with alum), while downregulated when injected with Pentavac SD (several antigens including hepatitis B, adjuvanted with whole-cell pertussis).Accelerating vaccine development: Insights gained from systems vaccinology can streamline the vaccine development process by identifying potential adjuvant candidates and optimizing their formulations for maximum efficacy. By providing detailed molecular data, this accelerates the screening and evaluation of adjuvants, reducing the time and cost associated with vaccine development. Moreover, these insights can guide the rational design of new adjuvants with specific properties, enhancing their ability to induce robust and long-lasting immune responses [48].Personalized vaccination strategies: Systems approaches help to personalize treatments and diagnostics by identifying differences between individuals or groups, giving preference to one treatment or diagnostic over another [49,50]. Similarly, by understanding how different populations respond at the molecular level, systems vaccinology can contribute to the development of personalized vaccination strategies, ensuring better protection across diverse groups. Transcriptomic analyses can reveal variations in gene expression responses to adjuvants among different demographic groups related to age, gender, and genetic background. This information can be used to tailor vaccines to specific populations, improving their efficacy and safety.

## 5. Investigating the Interactions Between Antigens and Adjuvants

Systems biology approaches have helped to reveal the details of pathogen–host integrations in several viral and bacterial diseases [51,52]. Dual transcriptomics simultaneously measures the gene expressions of the host and intracellular pathogens during an infection, which can be compared with their expression measurements prior to infection. Similarly, by profiling samples exposed to several permutations of treatment combinations, these techniques allow the prediction of the response to drug combinations in different diseases [53,54]. Since the effect the adjuvants have in the immune system depends on interactions with the antigen and other vaccine components, employing systems vaccinology helps us to understand the interactions between antigens and adjuvants through multi-condition experiments (antigen only, adjuvant only, and vaccine administration). Understanding the details of this interaction will help solve the major challenge of developing adjuvanted vaccines in which each adjuvant is only effective in certain formulations.

To observe the importance of understanding such interactions, Calabro and colleagues [13] measured mice gene expression when injected with a trivalent influenza vaccine adjuvanted with MF59 (which comprises citrate buffer, squalene emulsion, Span 85, and Tween 80), and compared the effects against mice injected with the same antigens but adjuvanted by either just the citrate buffer, just squalene, just Span 85, or just Tween 80. While the MF59-adjuvanted vaccine significantly affected the expression of ~400 genes (compared to antigens with PBS), less than half (exact numbers of DEGs were not provided) were affected when the antigens where adjuvanted by Span 85, and less than 10 genes when adjuvanted with any of the other ingredients. This demonstrates the level of detail achieved using transcriptional systems vaccinology approaches and how this results in the rational design and formulation of vaccines.

## 6. Use of Model Animals in Adjuvant Research

While studies on human subjects are generally restricted to the use of peripheral blood samples, these are not sufficient to inform on the mechanisms of adjuvants, as relevant events affected by vaccination are known to happen only locally near the injection site or in the nearby draining lymph nodes. A study that measured gene expression in the muscle, blood, and lymph nodes of mice after a vaccination adjuvanted by Poly I:C showed a great number of DE genes in the lymph nodes from 4 to 72 h, while DE genes in the muscle appeared mostly after 48 h, and in the blood there were almost no DE genes after 8 h [12].

A recent review on in vitro organoid models of human tissues shows how recently developed organoids can be used to model the interaction of several cell populations in the development of protective adaptive immunity [55]. But despite these recent advances, immune cell populations like monocytes and dendritic cells rapidly decline, making organoids unable to model any long-lasting effect. The review highlights that organoid models are not appropriate to study cell migration regulated by adjuvants.

Some animals, like non-human primates (NHPs), can be used as model animals instead of mice for reasons that include the need for a model with a longer lifespan, greater genetic variability, and differences in immune response [23]. One example of such murine–human differences that is particularly relevant for adjuvant research is the increased inflammatory response to lipopolysaccharides in humans due to differences in the regulation of TLR pathways [56]. However, many regulatory bodies, like the European Directorate—General Health and Consumer Protection, state that the use of NHPs for research should be avoided at all costs, and only used in cases where no other animal could be used as an alternative [57]. Despite differences in immune response, a careful meta-analysis of the genomic responses to systemic inflammation in human and murine models shows that genomic responses in mouse models greatly mimic human inflammatory diseases [58]. Humanized mouse models can be used to partially overcome human–mouse immune differences, like in the case of a recent study that showed that mice grafted with human tissues have a response to adjuvants PolyI:C and CpG-B closer to the human response than to that of normal mice [59].

## 7. Different Adjuvants Can Achieve Similar Improvement Through Different Pathways

Studies that involve two or more adjuvants are rarely able to properly compare them, as gene expression differences are confounded by the different antigens used for vaccine formulation in each study.

An interesting case was one in which rhesus macaques (*Macaca mulatta*) were vaccinated with several formulations where an HIV envelope was adsorbed to alum and TLR4 or TLR7 adjuvants. The results showed that several of these adjuvants promoted very similar increases in vaccine response [19]. Vaccines prepared with aluminium salts and either TLR4 or TLR7 agonists (Alum + TLR4, Alum + TLR7) both produced significantly more IgG than those adjuvanted by aluminium salts alone. Although precise DEGs were not shown, the gene expression obtained from the microarrays was clustered into 225 pre-defined gene functional modules. While all 34 modules annotated as “innate immune related” were significantly upregulated, either when comparing Alum with Alum + TLR4 or with Alum + TLR7, 15 of these were significantly upregulated when comparing Alum with Alum + TLR4 and downregulated when comparing Alum with Alum + TLR7. Modules with this switched regulation pattern include those annotated as platelet activation, inflammation, monocyte enrichment, and neutrophil enrichment, showing key differences in modes of action despite both adjuvants being combinations of aluminium salts and TLR agonists.

In another example involving TLR agonists, gene expression was measured in blood samples from healthy human donors treated with several TLR2, TLR3/MDA5, TLR4, TLR5/NLRC4, TLR7, TLR7/8, and TLR8 agonists [15]. The genes with the most distinct expression among the different treatments were CC and CXC chemokines, as well as interferons. Using the immunofluorescence detection of TLR1, 2, and 6 in foreskin fibroblasts that expressed TLR2 and 6, but not TLR1, the researchers observed an increase in the production of chemokines CCL20 and CXCL6 when treated with TLR2/6, but not TLR1/2 agonists, and that those same agonists were able to elicit a dose-dependent chemotaxis of PBMCs toward fibroblasts.

## 8. Challenges of Comparing Results of Different Studies

Vaccine adjuvants are cornerstones in vaccine design, formulation, and manufacturing, as they play a critical role in enhancing immune responses and improving vaccination efficacy. With an interest in better understanding the mechanism of action of vaccine adjuvants, we reviewed the literature for previous studies describing how vaccine adjuvants interact with the immune system, in the presence or absence of antigens, to boost immune system performance. In most cases, an adjuvant was found to enhance immune response, but adjuvant effects vary greatly depending on the paired antigen and form of delivery.

In [18], mice vaccinated against rabies had better long-term IgG production when adjuvanted with Alum than with MF59, while [17] showed that mice vaccinated against H1N1 had better long-term IgG production when adjuvanted with MF59 than with Alum. These differences in results suggest that it is not possible to directly compare adjuvant effects when the antigens are not conserved [12]. So far, in the works that we surveyed, the rationale of selecting which adjuvant would be a reasonable pair to an antigen was unclear, or it was simply based on an adjuvant that worked in previous vaccines for evolutionary-related organisms.

In very few cases, different adjuvants were compared in the same study, but it is challenging to make any generalizations on their properties due to differences specific to each experimental setting. These differences in antigens, hosts, dosage, and the location and timing of sampling, among others, make the adjuvant impacts significantly different between studies (Figure 1A). This lack of consistency makes it challenging to perform a meta-analysis on adjuvant research with the available data, as there are no common features to be used as a control for batch effects or as other confounding variables. These disparities hinder the prediction of patterns necessary for scientific reasoning.

Furthermore, different cell types showed different responses to diseases and vaccines. We recently showed that the variant behaviour of disease in genes can only be captured on the single-cell level [60]. Only a few studies from our survey, such as [15,24,29], show how differently each cell type is affected by adjuvanted vaccines, and how important cytokine-mediated communication is between these cells. This, therefore, reinforces the need for further works that utilize modern single-cell (sc) techniques (i.e., scRNA-Seq and scMS-based proteomics) and take cell-type specificity into account for accurate systems vaccinology modeling.

## 9. Future Directions

Many different mechanisms of action are used by each antigen to achieve antigen-specific antibody-mediated responses [12,19,20]. However, the available data and studies cannot adequately explain the details of these mechanisms or help in deciding which mechanism is used in different contexts. Therefore, more full-system transcriptional vaccinology profiling studies are necessary to observe the details of the mechanisms of action at the molecular level, such as which pathways are necessarily involved, and which are the downstream genes that need to be targeted to achieve lasting immune response (Figure 1B). Such studies would provide a better understanding of how adjuvants work and support the rational selection of adjuvants in the vaccine formulation process (Figure 2).

## 10. Conclusions

In conclusion, we must stress that the study of adjuvants in vaccine development is still lacking the mechanistic understanding that allows for a proper adjuvant classification and rational selection of adjuvants for vaccine formulation. Achieving this understanding requires more unified experimental designs profiling several adjuvants under similar conditions, as well as granular transcriptomic data (in bulk and single-cell levels) showing how every gene and immune pathway interact with the adjuvant and/or antigen in different cell types and over time, from antigen presentation to the differentiation of memory cells.

## Figures and Tables

**Figure 1 vaccines-13-00033-f001:**
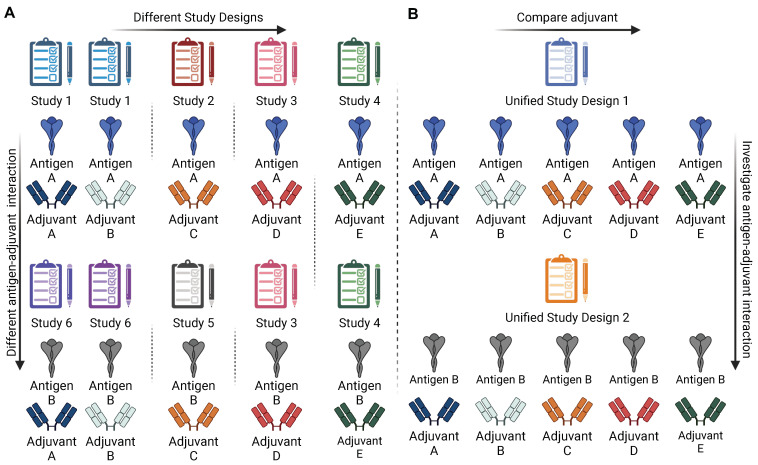
(**A**) The available data on adjuvant profiles and antigen–adjuvant interaction studies are from experiments where adjuvants tested under different conditions create confounding factors that make it impossible to make comparisons or create an accurate profile. (**B**) Suggested transcriptional systems vaccinology approach for adjuvant profiling.

**Figure 2 vaccines-13-00033-f002:**
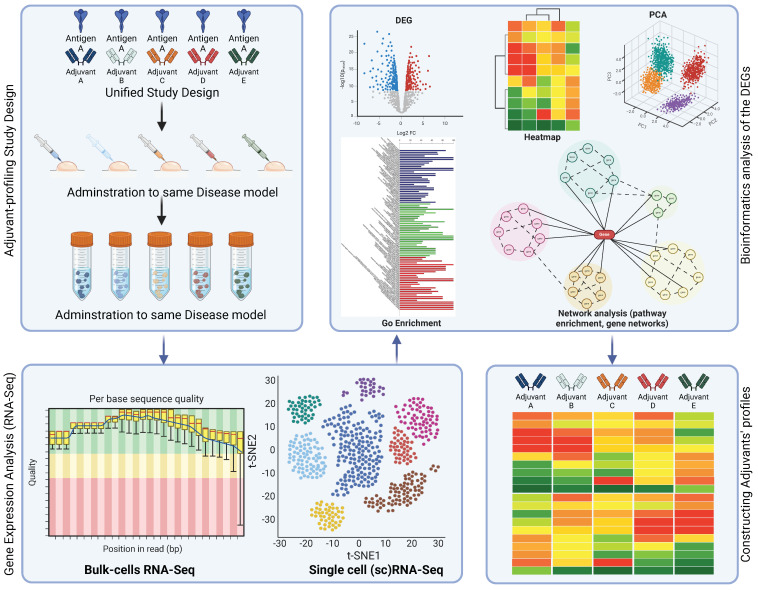
Suggested experimental design: A unified experimental design can create controlled conditions and patterns can be inferred on the adjuvants’ effects. Predicting which vaccine formulations will work effectively with a given adjuvant remains a significant challenge. The efficacy of an adjuvant is highly dependent on its interactions with the antigen and other molecules, as demonstrated in [13]. Future vaccine development could be made simpler if a better understanding of these interactions is available for scientists so that they can focus on testing the more promising adjuvants.

**Table 1 vaccines-13-00033-t001:** An overview of studies using systems vaccinology approaches to investigate adjuvant mechanisms of action.

Study	Experimental Method	Adjuvants	Organism	Antigen	Immune Challenge	Publicly Available Data
[15]	96-well PCR	17 types of TLR agonists (2, 7, 7/8, 8)	humans	no antigen	no	no
[16]	microarray	AddaVax, CpG K3, Poly I:C, Al, DMXaa, R484, NanoSiO2, and Pam3CSK4	mice	H1N1	no	no
[12]	microarray	MF59, LPS, pentavac, poly I:C, and IFA	mice	several vaccines	no	GSE120661
[17]	ELISA	Alum, MF59, GLA-SE, IC31, and CAF01	mice	tuberculosis, chlamydia, influenza	no	no
[18]	arrays	bacterial-like particles, AS02, AS03, MF59, and Poly I:C	mice	rabies	rabies	no
[19]	microarray	Alum, TLR4 and 7 agonists, MF59, and pIC:LC	rhesus	HIV	no	no
[20]	microarray	CAF01, IC31, GLA-SE and Alum	mice	H56 protein	no	GSE85339
[21]	arrays	AS01B, AS01E, and AS03	humans	hepatitis B	no	GSE116975
[22]	single-cell qPCR	CNE and MF59	mice	H1N1	no	no
[23]	microarray	Advax1 and LNP	rhesus	cytomegalo-virus	no	no
[24]	RNA-Seq	Poly (I:C) and ISA 720	mice	malaria	no	GSE203218
[25]	antibody array	AS03 and MF59	humans	influenza A	no	GSE202392
[26]	ELISA	AH and CAF01	mice	chlamydia, tuberculosis	no	no
[27]	ScRNA-Seq	CD8α AcTaleukin-1/ALN-1	mice	influenza	H3N2	GSE134647
[28]	scRNA-Seq, scATAC-Seq	AS03	humans	H5N1	ex vivo dengue and zika	GSE166063, GSE102012, GSE165907
[29]	scRNA-seq, ATAC-seq	3M-052	mice	yellow fever live attenuated virus	no	GSE180384
[7]	RNA-Seq	ZIF-8	mice	COVID-19	no	GSE246178
[30]	RNA-Seq	MDP/R848	mice	OVA-NCs	no	no
[31]	RNA-Seq	CAF01	mice	tuberculosis	repeated tuberculosis antigen	PRJNA437839
[32]	RNA-Seq	Alum	calves	tick saliva	ticks	DEG Table
[33]	RNA-Seq	Alum	sheep	several vaccines	no	DEG Table
[34]	RNA-Seq	Alhydrogel	sheep	several vaccines	no	DEG Table
[13]	microarray	MF59	mice	influenza	no	E-MTAB-1408

Abbreviations: PCR: polymerase chain reaction; TLR: toll-like receptor; CpG: Cytosine phosphate Guanine; Alum: aluminium salts; DMXaa: Dimethylxanthenone acetic acid; R848: Resiquimod; NanoSiO2: nanoparticles of silica dioxide; Pam3CSK4: Pam3CysSerLys4; MF59: microfluidized emulsion 59; LPS: lipopolysaccharide; poly I:C: Polyinosinic–polycytidylic acid; IFA: Incomplete Freund’s Adjuvant; Fluad: influenza adjuvant; GLA-SE: Glucopyranosyl lipid adjuvant-stable emulsion; IC31: Intercell 31; CAF01: Cationic Adjuvant Formulation 01; AS02: Adjuvant System 02; AS03: Adjuvant System 03; pIC:LC: polyinosinic–polycytidylic acid–poly-L-lysine; carboxymethylcellulose; H56: hybrid 56; AS01: Adjuvant System 01; qPCR: real-time polymerase chain reaction; CNE: cationic nano emulsion; LNP: lipid nanoparticles; ISA 720: Incomplete Seppic Adjuvant 720; AH: aluminum hydroxide; ALN-1: Alendronate 1; ZIF-8: zeolitic imidazolate framework–8; MDP/R848: muramyl dipeptide (MDP) and resiquimod; OVA-NCs: ovalbumin-based nanocapsules; DEG: differentially expressed genes; 3M-052L: Telratolimod. Addavax1 is a commercial name of an adjuvant.

**Table 2 vaccines-13-00033-t002:** Adjuvanted FDA-approved vaccines.

Vaccine	Adjuvant	Adjuvant Group
GARDASIL	Amorphous aluminium hydroxyphosphate sulfate	Aluminium salts
GARDASIL 9		
PedvaxHIB		
RECOMBIVAX HB		
VAQTA		
Adacel	Aluminium phosphate	
DAPTACEL		
PENBRAYA		
Pentacel		
Prevnar 13		
Prevnar 20		
Quadracel		
TDVAX		
TENIVAC		
VAXELIS		
VAXNEUVANCE		
PEDIARIX	Aluminium hydroxide and aluminium phosphate	
TWINRIX		
Bexsero *	Aluminium hydroxide	
BioThrax		
BOOSTRIX		
HAVRIX		
INFANRIX		
IXIARO		
KINRIX		
PREHEVBRIO		
TICOVAC		
CERVARIX	Aluminium hydroxide and MPL (AS04)	Oils
AREXVY	MPL and QS-21 (AS01B)	
SHINGRIX		
Arepanrix H5N1	Squalene, DL-α-tocopherol, and polysorbate 80 (AS03)	
AUDENZ	Squalene, polysorbate 80, and sorbitan trioleate (MF59)	
Fluad		
Fluad Quadrivalent		
HEPLISAV-B	CpG oligodeoxynucleotides 1018	CpG oligodeoxynucleotides
CYFENDUS	CpG oligodeoxynucleotides 7909 and aluminium hydroxide	
ERVEBO	Recombinant vesicular stomatitis virus (rVSV)	Viral vector
Comirnaty	RNA sequences in mixed lipid nanoparticles	Adjuvant for RNA vaccines **
Spikevax		

* Bexsero includes outer membrane vesicles, but they were added as antigens. ** These adjuvants are specific to RNA vaccines that do not include the antigen as a protein. Exogenous RNA has its own immunostimulatory proprieties and lipid vesicles have a role in transporting this RNA sequence into the cytoplasm where it will be translated into the antigen [40].

## Data Availability

Not applicable.

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
