# Peer review of "Transcriptional Systems Vaccinology Approaches for Vaccine Adjuvant Profiling"

_vaccines, 2025, doi:10.3390/vaccines13010033_

Round 1

Reviewer 1 Report

Comments and Suggestions for Authors

In the manuscript entitled “Transcriptional Systems Vaccinology Approaches for Vaccine Adjuvant Profiling” the authors discuss the utility of transcriptional systems vaccinology strategies for the comprehensive characterization of adjuvants. Below are the comments to improve the manuscript.

1.      The authors should explain the detailed rationale that was employed to select the studies for this manuscript.

2.      Did the authors perform meta-analysis of the publicly available data sets from the enlisted studies in the manuscript? This approach may reveal novel insights.

3.      Wherever practically possible we can use human organoids/humanized mouse models/HLA transgenic mice/outbred mouse models to simulate human diversity.  Did the authors come across such studies? The authors should incorporate these facets in the discussion.

Author Response

1. The authors should explain the detailed rationale that was employed to select the studies for this 
manuscript.
R1. We added a paragraph to the end of the introduction, to explain how we selected those 
papers. Section 1, last paragraph.

2. Did the authors perform meta-analysis of the publicly available data sets from the enlisted 
studies in the manuscript? This approach may reveal novel insights.
R2. It is challenging to compare these papers in a meta-analysis because of the lack of 
common variables between papers. We tried selecting samples that were publicly available 
for a meta-analysis, but very few of those were comparable. We updated the third paragraph 
of section 8 to further reflect that.

3. Wherever practically possible we can use human organoids/humanized mouse models/HLA 
transgenic mice/outbred mouse models to simulate human diversity. Did the authors come across 
such studies? The authors should incorporate these facets in the discussion.
R3. None of the papers we citied previously used these models, but we added a new 
paragraph to section “6”, and extended the last paragraph of the same section, citing two 
examples and discussing those models in more depth. Section 6, 2nd and last paragraphs.

Reviewer 2 Report

Comments and Suggestions for Authors

Adjuvants plays an critical role in enhancing the body's immune response to the co-administered antigen, which itself is not sufficiently immunogenic to elicit optimal immune responses. They are especially critical for subunit vaccines as these proteins alone might not be recognized as a threat by the immune system. Despite the importance and wide-spread usage of the adjuvants in vaccine formulations, their mechanisms of action are still not fully understood. The the manuscript entitled ‘Transcriptional Systems Vaccinology Approaches for Vaccine Adjuvants Profiling’, Diogo et. al. underscore the significance of transcriptomics in symtems vaccinololgy approaches. After reviewing how vaccine adjuvants interact with the immune system when administered alone or in combination with antigens, they focused on reviewing the application of transcriptomics in analyzing the mechanism of adjuvant action in systematic vaccine approaches, in order to assist in the rational selection of adjuvants during vaccine formulation. The review is well-written and informative.

With the rapid development of artificial intelligence (AI) technology, AI has been applied to many fields. A question not mentioned in the article is whether AI is applied in the field of adjuvant selection? For example, is there an AI model that can predict the possible mechanism of action of adjuvants, which adjuvants are more suitable for certain animals, antigens, or immune pathways. Please try to discuss.

Author Response

1. With the rapid development of artificial intelligence (AI) technology, AI has been applied to 
many fields. A question not mentioned in the article is whether AI is applied in the field of adjuvant 
selection? For example, is there an AI model that can predict the possible mechanism of action of 
adjuvants, which adjuvants are more suitable for certain animals, antigens, or immune pathways. 
Please try to discuss.
R1. The construction of an AI model such as a neural network to do a meta-analysis of the 
interaction of adjuvants in different animals, antigens, or immune pathways is not currently 
feasible, and that is in line with our main conclusions. However, since AI models require data 
for training and testing, the available data are made in very heterogeneous settings, and 
without common controls so that it is almost impossible to normalize, or batch correct the 
data in order to make samples from different sources comparable at all. Hence, creating 
robust AI models to help in adjuvant profiling or selection requires the type of data that we 
recommended in our conclusions. 
We updated the third paragraph of section “8” to discuss this more. Our “Future Directions”
section calls for more comprehensive studies with controls to make these comparisons 
possible and make the data more integrable and interoperable.

Reviewer 3 Report

Comments and Suggestions for Authors

I have carefully read this perspective manuscript and find it both interesting and worthy of publication. However, I have several comments for the authors to consider, which I believe will improve the quality and clarity of the manuscript.

Historical Reference (Lines 45–47):
The authors state:
“For almost a century, adjuvants have played a key role in improving the efficacy of vaccines by stimulating the immune system to respond more vigorously to vaccine antigens.”
I recommend being more specific and including the historical reference for the first use of an adjuvant in vaccines. This would provide greater context and strengthen the statement.

Tables Format and Content:
Tables should be formatted in landscape orientation and must present information that is fully comprehensible without requiring reference to the main text. I suggest defining all acronyms and abbreviations in footnotes directly under each table. There is significant room for improvement in the format and organization of the table content to enhance its readability and clarity.

Clarity of Paragraph (Lines 67–74):
The paragraph in lines 67–74 is confusing and should be rewritten. Shorter sentences would improve readability and flow.

Formatting of Scientific Names (Line 306):
Ensure that the name of the species mentioned in line 306 is italicized, following standard conventions.

Conclusions and Future Directions Section:
The section titled "Conclusions and Future Directions" is not appropriately structured. Conclusions should be concise and reflect the authors’ opinions based on the reviewed literature. The current section includes references and data from other studies, which should instead appear in the main body of the manuscript.

The authors present a figure in this section, which is not appropriate here. This figure should be enlarged for better readability and moved to a more suitable section of the manuscript.

In the conclusions, the authors should emphasize their vision of how Transcriptional Systems Vaccinology Approaches could be useful for Vaccine Adjuvant Profiling.

Citation Formatting:
Please review the citation formatting to ensure compliance with the journal’s style guide, as it currently does not adhere to the required format for Vaccines.

Author Response

1. Historical Reference (Lines 45–47):
The authors stated:
“For almost a century, adjuvants have played a key role in improving the efficacy of vaccines by 
stimulating the immune system to respond more vigorously to vaccine antigens.”
I recommend being more specific and including the historical reference for the first use of an 
adjuvant in vaccines. This would provide greater context and strengthen the statement.
R1. We updated our first paragraph of section “1” to talk about the adjuvant developments
and the gap from 1930 to 2009. More recent developments are discussed throughout the text.

2. Tables Format and Content:
Tables should be formatted in landscape orientation and must present information that is fully 
comprehensible without requiring reference to the main text. I suggest defining all acronyms and 
abbreviations in footnotes directly under each table. There is significant room for improvement in 
the format and organization of the table content to enhance its readability and clarity.
R2. Since our tables have more raws than columns, placing them in landscape was very 
detrimental to its readability. We added abbreviations to the foot note of table 1. Also, we 
made some modifications to replace acronyms for their full name in the tables, whenever 
possible. We also expect that these tables would be improved in the final editing stages, 
following the journal style.

3. Clarity of Paragraph (Lines 67–74):
The paragraph in lines 67–74 is confusing and should be rewritten. Shorter sentences would 
improve readability and flow.
R3. The third paragraph of section 1 was reviewed and should be clearer to understand now.

4. Formatting of Scientific Names (Line 306):
Ensure that the name of the species mentioned in line 306 is italicized, following standard 
conventions.
R4. The scientific name of the Rhesus macaque was correctly formatted on the second 
paragraph of section 7. No other instances of scientific names were found.

5. Conclusions and Future Directions Section:
The section titled "Conclusions and Future Directions" is not appropriately structured. Conclusions 
should be concise and reflect the authors’ opinions based on the reviewed literature. The current 
section includes references and data from other studies, which should instead appear in the main 
body of the manuscript.
R5. We have re-structured this part of the manuscript. The old conclusion section is now 
updated and became section “8. Challenges of comparing results of different studies” and 
we added a separate conclusions section, the new section 10.

6. The authors present a figure in this section, which is not appropriate here. This figure should be 
enlarged for better readability and moved to a more suitable section of the manuscript.
R6. The figure was split to two figures and the font was increased wherever possible. They 
are now cited in the new section 8. Challenges of comparing results of different studies”.

7. In the conclusions, the authors should emphasize their vision of how Transcriptional Systems 
Vaccinology Approaches could be useful for Vaccine Adjuvant Profiling.
R7. The restructuring of the old conclusions section and the newly added concise conclusion 
section “section 10” are now more focused on adjuvant profiling.

8. Citation Formatting:
Please review the citation formatting to ensure compliance with the journal’s style guide, as it 
currently does not adhere to the required format for Vaccines.
R8. The citation style was changed to the numerical format.